# Structural and regulatory determinants of flagellar motility in *Rhodobacterales*—the archetypal flagellum of *Phaeobacter inhibens* DSM 17395

Jürgen Tomasch,[1] Pascal Bartling,[2] John Vollmers,[3] Lars Wöhlbrand,[4] Michael Jarek,[5] Manfred Rohde,[5] Henner Brinkmann,[2] Heike M. Freese,[2] Ralf Rabus,[4] Jörn Petersen[2,6]

**ABSTRACT**   Flagellar motility is crucial for the swim-and-stick lifestyle and plays an important role in bacterial-algal interactions of *Rhodobacterales*. This alphaproteobacterial order contains three distinct types of flagellar gene clusters (FGCs) for the formation of a functional flagellum. Our phylogenetically broad taxon sampling of more than 300 genomes revealed that the most common FGC, the *fla1*-type, was probably already present in the common ancestor of *Rhodobacterales* and was strictly vertically inherited, while the other two FGC types, *fla*2 and *fla*3, were spread via horizontal operon transfers. Swimming of the marine model organism *Phaeobacter inhibens* DSM 17395 (*Roseobacteraceae*) is mediated by the archetypal *fla1*-type flagellum. Screening of 13,000 transposon mutants of *P. inhibens* on soft agar plates revealed that 40 genes, including four genes encoding conserved but not yet characterized proteins (CP1–4) within the FGC, are essential for motility. Exoproteome analyses indicated that CP1–4 are required at different stages of flagellar assembly. Only eight genes outside the FGC were identified as essential for swimming motility, including all three genes of the CtrA phosphorelay. Using comparative transcriptomics of Δ*cckA*, Δ*chpT*, and Δ*ctrA* mutants of the distantly related model organisms *P. inhibens* and *Dinoroseobacter shibae* DSM 16493, we identified genes for the flagellum and cyclic di-GMP turnover as core targets of the CtrA phosphorelay and a conserved connection with quorum sensing across members of the *Rhodobacterales*.

**IMPORTANCE**   The bacterial flagellum is a sophisticated nanomachine for swimming motility and rapid chemotactic response to gradients of attractants or repellents in the environment. It is structurally highly conserved and has been intensively studied in gammaproteobacterial model bacteria such as *Escherichia coli* and *Salmonella enterica*. However, the flagellar gene clusters of different alphaproteobacterial orders have distinct structures and compositions, as demonstrated by the three flagellar systems of *Rhodobacterales* investigated in the current study. The archetypal *fla1*-type flagellum originated in its common ancestor and evolved synchronously with the host. The universal presence of four as yet uncharacterized essential genes in *fla1*-type FGCs (CP1–4) reflects the order-specific adaptation of the flagellar system during bacterial evolution. Comparative transcriptome analyses of Δ*cckA*, Δ*chpT,* and Δ*ctrA* mutants showed that the core function of the CtrA phosphorelay in *Rhodobacterales* is the transcriptional control of flagellar genes.

**KEYWORDS**   flagellar motility, flagellar gene regulation, CtrA phosphorelay, *Rhodobacterales*, evolution

**Peer Reviewer** Elias Javier Mongiardini, CONICET La Plata, La Plata, Argentina

Address correspondence to Jürgen Tomasch, tomasch@alga.cz, or Jörn Petersen, joern.petersen@dsmz.de.

Jürgen Tomasch and Pascal Bartling contributed equally to this article. Author order was determined by the contribution to the writing of the manuscript.

The authors declare no conflict of interest.

See the funding table on p. 17.

The bacterial order *Rhodobacterales* (phylum *Pseudomonadota*, class *Alphaproteobacteria*) contains two major families, namely the *Paracoccaceae* and *Roseobacteraceae* (1, 2). While some species are found in soil or freshwater (3), the *Roseobacteraceae* are an abundant and important part of the marine microbial community. They are found in the phycosphere of macro- and microalgae (4–9), where they contribute to both primary production and remineralization of nutrients (10–12). In model co-culture systems, they promote growth but can also induce the death of their algal hosts (5, 13–15). In several cases, motility was found to be essential for interaction with the eukaryote (16–18). The ability to switch between growth in biofilms on biotic or abiotic surfaces and as free-roaming pelagic cells is widely conserved within this family (19, 20) and is likely a key factor for their success in the environment (21, 22). This swim-or-stick lifestyle of the *Roseobacteraceae* resembles the biphasic growth of *Caulobacter vibrioides* (synonym *C. crescentus*). In this alphaproteobacterial model organism, decades of work have revealed a complex regulatory cascade that coordinates flagellar gene expression with the cell cycle (23). At its center is the CtrA phosphorelay, which likely evolved from an ancient regulator of flagellar gene expression and is conserved in *Alphaproteobacteria*, including the *Rhodobacterales* (24).

The bacterial flagellum is a complex nanomachine that is assembled sequentially, starting from a motor that is anchored in the membrane(s) and ending in the filament composed of numerous subunits of flagellin (25, 26). Its rotation is typically powered by a proton gradient mediated by stator proteins that form a proton channel (25). The flagellum is crucial for swimming motility (27), and its rotation is often regulated by chemotaxis proteins in response to attractants or repellents (28). About 35 different proteins are involved in flagellar assembly, with some of them acting as chaperones and not being part of the final structure. A set of 24 core proteins is largely conserved across bacterial phyla (26). Flagellar genes in *Rhodobacterales* are also highly conserved and typically located in a flagellar gene cluster (FGC) (29, 30), which probably facilitates their coordinated expression (26, 31). A decade ago, a comparative genome analysis of 40 taxa revealed the presence of three evolutionary distinct FGCs (*fla1*, *fla2*, and *fla3*) in *Rhodobacterales* (30). Each of them alone can confer the ability to swim on soft agar plates (20). Notably, some members of *Rhodobacterales* contain more than one FGC encoded in their genomes. Some of the *fla2*- and *fla3*-type FGCs were found on plasmids, suggesting that they have been distributed across species borders by horizontal operon transfer (HOT).

The present study aimed to provide a comprehensive overview of the distribution and evolution of flagellar systems in the *Rhodobacterales*. *Phaeobacter inhibens* DSM 17395, a probiotic bacterium in marine aquacultures (32) and an important model organism for algal-bacterial interactions (5, 15, 17), heterotrophic C/N-cycling (33, 34), and plasmid biology (14, 35–37), was chosen to determine structural and regulatory components of flagellar motility. Screening of a transposon mutagenesis library revealed 40 genes that are essential for motility, and we characterized mutants of conserved structural proteins within the FGC and the CtrA phosphorelay using proteomics and transcriptomics, respectively.

## RESULTS

### Phylogenomic tree of 306 *Rhodobacterales*

The prerequisite for reliable insights into flagellar evolution was the establishment of a robust phylogenomic reference tree. Accordingly, the selection of 306 *Rhodobacterales* genomes was focused on a phylogenetically broad taxon sampling (Table S1). It comprised 230 type strains, but also five strains previously used for the investigation of flagellar motility: *Phaeobacter inhibens* DSM 17395, *Epibacterium* sp. TM1040 (also *Silicibacter* sp. TM1040 or *Ruegeria* sp. TM1040), *Marinovum algicola* DG898, *Dinoroseobacter shibae* DSM 16493, and *Cereibacter sphaeroides* 2.4.1 (synonym *Rhodobacter sphaeroides* 2.4.1). The phylogenomic tree that is presented in Fig. 1A was rooted with *Rubrimonas cliftonensis* DSM 15345, representing the most basally branching bacterium

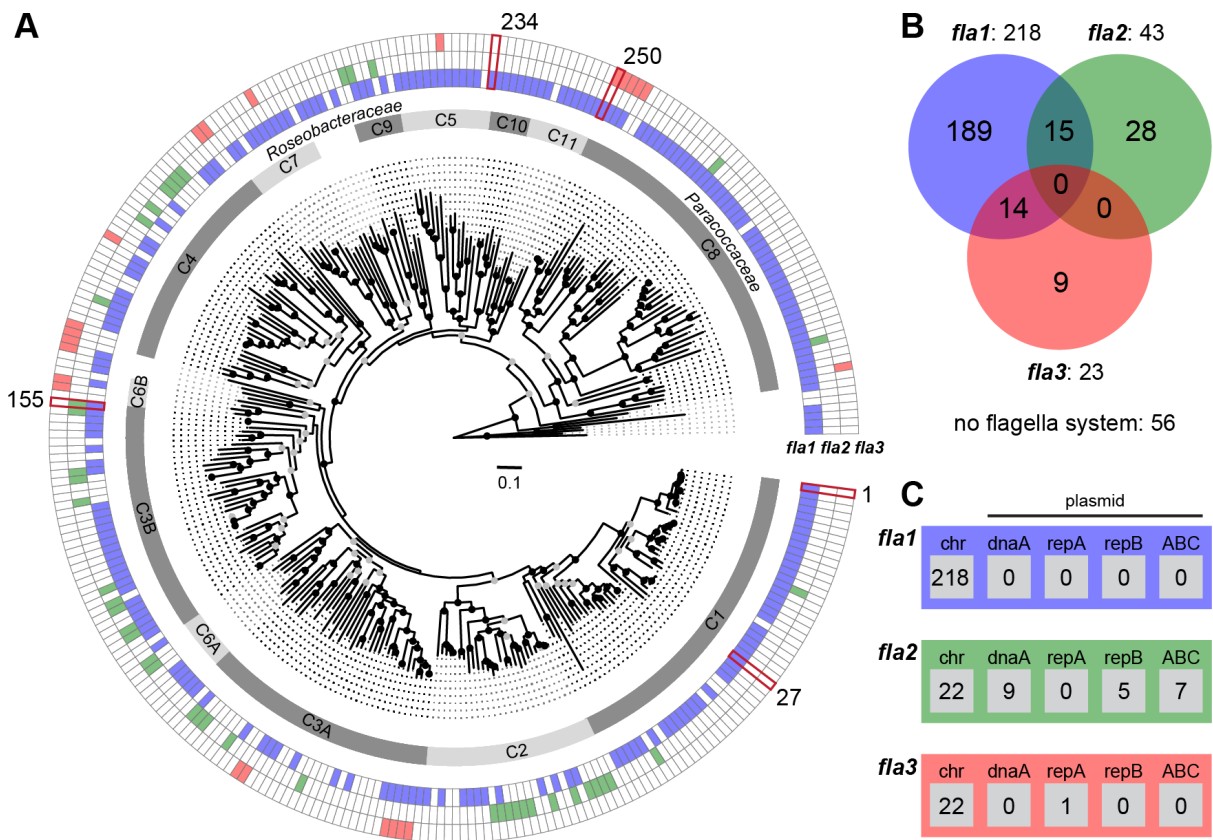

**FIG 1** Phylogenetic distribution of three different flagella gene clusters in *Rhodobacterales*. (A) Phylogenomic tree of 306 *Rhodobacterales* strains based on an alignment of 123,793 amino acid positions from 417 orthologous proteins. Bootstrap support is indicated as black (100%) or gray (> 70%) circles. The scale bar represents amino acid substitutions per site. The fully annotated tree is shown in Fig. S1A; genome accession numbers are listed in Table S1. The inner ring shows the 11 distinct clades of *Rhodobacterales* (C1–C11). The outer rings indicate the presence and absence of FGCs corresponding to the three different flagellar systems, namely *fla1* (blue), *fla2* (green), and *fla3* (red; Fig. S1B). (B) Venn diagram of the FGC abundance in the 306 genomes. (C) Genomic localization of FGCs on the chromosome (chr) and four plasmid types (DnaA-like [*dnaA*], RepA [*repA*], RepB [*repB*], and RepABC [ABC]) specific to *Rhodobacterales* is indicated in the respective boxes. Five reference strains for the study of flagellar motility are highlighted in dark red: *Phaeobacter inhibens* DSM 17395 (1), *Epibacterium* sp. TM 1040 (27), *Marinovum algicola* DG898 (155), *Dinoroseobacter shibae* DSM 16493 (234), and *Cereibacter sphaeroides* 2.4.1 (250).

of our data set (1). The phylogeny is well-resolved, and its branching pattern is supported by a 100% bootstrap proportion for many subtrees (BP, Fig. S1A). Identification and naming of different clades (C1–C11), which allowed a subclassification of taxa between family and genus level, is consistent with the results of previous studies (20, 38). The clades, ranging from *P. inhibens* DSM 17395 (clade 1) to *Rhodovulum sulfidophilum* DSM 1374 (clade 11), all represent the marine *Roseobacteraceae*, whose monophyly is supported by 92% BP, while clade 8 represents the distinct sister family *Paracoccaceae* (100% BP). Serial branching of the taxa ranging from *Meinhardsimonia xiamenensis* DSM 24422 to *Ru. cliftonensis* DSM 15345 indicated the presence of at least four as yet undescribed basal families in the order *Rhodobacterales*.

## Distribution and evolution of flagellar systems in *Rhodobacterales*

The distribution of flagellar systems in 306 *Rhodobacterales* genomes was investigated by comprehensive BLASTP searches using the three previously characterized FGCs from *P. inhibens* DSM 17395 (*fla1*), *M. algicola* DG898 (*fla2*), and *C. sphaeroides* 2.4.1 (*fla3*) as references (30). We identified 284 FGCs in 255 strains with a proportion of 77% flagellar systems of the *fla1*-type (218), 15% of the *fla2*-type (43), and 8% of the *fla3*-type (23; Fig. 1B). In total, 51 taxa harbored no FGCs, and two different flagellar systems were identified in 29 genomes. While 15 strains contained the combination *fla1*/*fla2* and 14

strains contained the combination *fla1*/*fla3*, neither the combination *fla2*/*fla3* nor the presence of all three flagellar systems was found.

Phylogenies of the *fla1*, *fla2*, and *fla3* FGCs were individually calculated based on four conserved proteins: FlhA, FliF, FlgH, and FlgI (30). A branching pattern comparable to the species tree was only found for the *fla1*-type flagellum, as illustrated by the formation of distinct flagellar subtrees reflecting the different clades of *Rhodobacterales* (Fig. S1B). The *fla1* subtree topology of clade 1 precisely mirrors the organismal evolution, and the absence of *fla1* in only 7 out of 50 genomes is therefore best explained by independent losses of the FGC. A comparable pattern of sporadic *fla1* losses could also be observed in other well-sampled lineages, such as clade 3B (8/35) or clade 8 (3/50) (Table S1). In contrast to the archetypal flagellar system *fla1*, neither the *fla2*- nor the *fla3*-phylogeny showed conserved subtrees beyond the genus level (Fig. S1B and S2).

Co-localization analysis of FGCs and plasmid replication systems revealed 22 FGCs located on extrachromosomal replicons (Fig. 1C). All *fla1*-type and all but one *fla3*-type FGCs were found to be located on the chromosome. By contrast, almost half (21/43) of the *fla2*-type FGCs are localized on plasmids representing three different replication types, namely DnaA-like, RepB, and RepABC, and in total nine compatibility groups (Fig. S2). The scattered distribution of *fla2*-type FGCs throughout 8 *Rhodobacterales* clades illustrates the promiscuity of this flagellar type (Fig. 1A). The corresponding flagellar tree showed that taxa of clade 3B recruited *fla2* at least three times independently (Fig. S2). The phylogenetic distance between chromosomally and mainly plasmid-encoded *fla2*-type FGCs within the genus *Sulfitobacter* (clade 2) reflects two independent ancient acquisitions, and the comparison of flagellar and species trees revealed two recent horizontal transfers of the *fla2*-type FGC for *S. marinus* DSM 23422 and *S. brevis* DSM 11443. In summary, the diverse compatibility groups of *fla2*-bearing plasmids and their phylogenetic distribution suggest a dominant role of plasmid-mediated HOTs in the evolution of *Rhodobacterales*.

## Structural composition of the archetypal flagellar system *fla1* of *Rhodobacterales*

A comparison of the archetypal *fla1*-type FGC from five well-studied *Rhodobacterales* strains demonstrated their structural conservation irrespective of their presence in *Roseobacteraceae* or *Paracoccaceae* (Fig. 2A). The upstream cluster of 29 genes ranging from *motB* to *fliI* (genes 01-29, Table S2A) showed a conserved genetic architecture with three exceptions. First, non-flagellar genes were inserted between the inversely oriented operons *motBflgEKLI* and *fliFHNP* in *D. shibae* (108 genes) and *C. sphaeroides* (two genes). Second, the conserved gene 19 with unknown function was replaced by a non-homologous gene in *D. shibae*. Third, the soluble lytic murein transglycolase (gene 15) was found exclusively in *P. inhibens* and *C. sphaeroides*. The last observation was of particular interest because the corresponding transposon mutant of *P. inhibens* was immotile (see below). The downstream cluster of seven genes, ranging from *flbT* to *flgD* (genes 30–36), underwent more frequent rearrangements, as documented by the distinct localization of the conserved protein CP1 in *P. inhibens* (Table S2A). It is noteworthy that *fla1* genes of the four *Roseobacteraceae* in Fig. 2A are generally located on the chromosome, whereas the conserved *fliC-flaF-flbT-CP1* gene cluster of *C. sphaeroides* was found on a 124 kb plasmid, reflecting inter-replicon recombination. This plasmid also contained a conserved *rmlCBDA* operon for the biosynthesis of L-rhamnose that is crucial for biofilm formation of *Rhodobacterales* (19), thus documenting a genetic linkage of surface attachment and flagellar motility.

Comparison of *fla1* of *Rhodobacterales* with the FGC of *Agrobacterium fabrum* C58 (*Hyphomicrobiales*) and *Rhodospirillum rubrum* S1 (*Rhodospirillales*) revealed fundamental differences in their composition (Fig. 2B and C). In addition, chemotaxis genes were part of the FGCs of *A. fabrum* and *Rh. rubrum* (Table S2B and C), which was not observed for the FGCs of *Rhodobacterales*. Furthermore, the three genes of the CtrA phosphorelay (*ctrA*, *cckA*, and *chpT*) were associated with flagellar genes in *Rh. rubrum*. The most

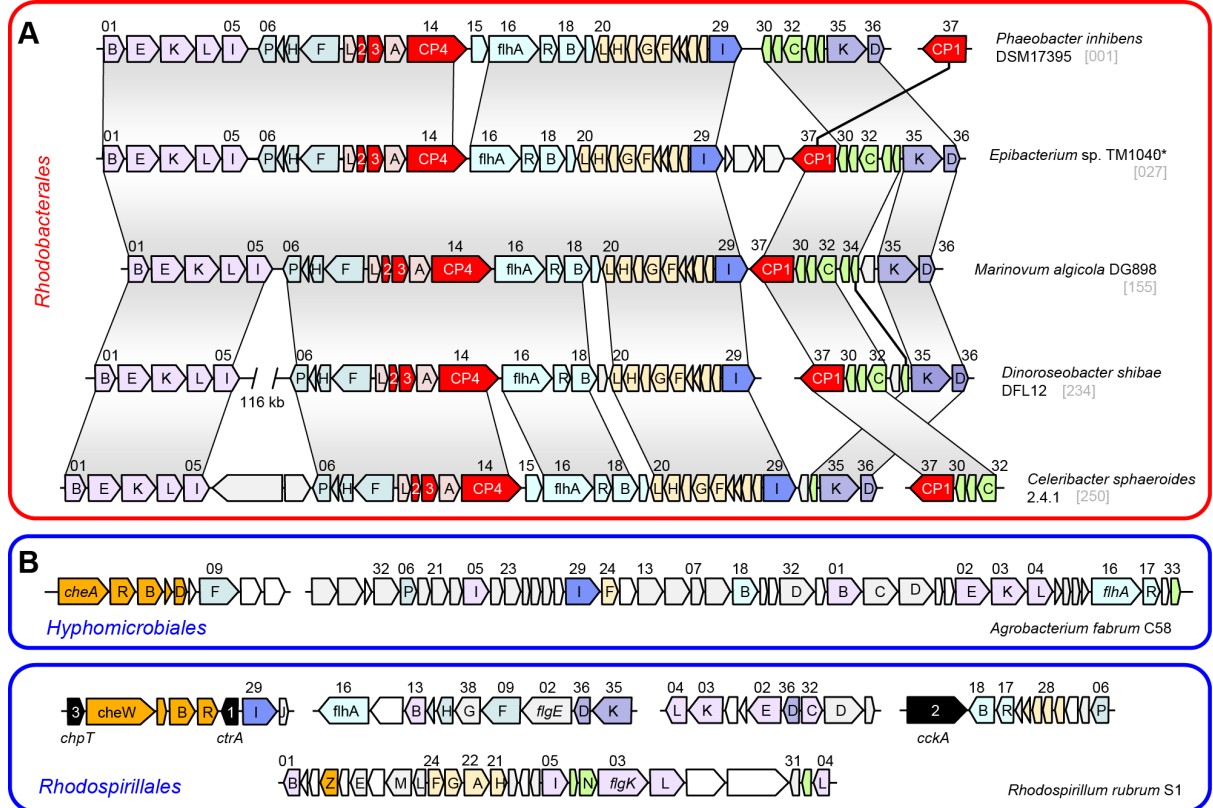

**FIG 2** Organization of flagellar gene clusters in *Alphaproteobacteria*. (A) *Fla1*-type FGCs of five representative *Rhodobacterales*. Genes coding for four conserved hypothetical proteins are highlighted in red. (B) FGCs of representatives of *Hyphomicrobiales* and *Rhodospirillales*. The flagellar genes are numbered according to their position in the FGC of *P. inhibens* DSM 17395 (Table S2A). Chemotaxis genes are shown in orange; *also *Silicibacter* sp. TM1040.

peculiar finding of our study was the universal presence of four conserved proteins designated CP1–4, which were exclusively found in *fla1*-, but not in *fla2*- and *fla3*-type FGCs of *Rhodobacterales* or outside this order (Fig. 2). All four proteins are essential for flagellar motility of *P. inhibens* DSM 17395 (see below) and were apparently recruited by a common ancestor of *Rhodobacterales* concomitant with the evolutionary emergence of a functional *fla1*-type flagellum.

## Motility screening of a genome-wide *P. inhibens* transposon library

Out of the five completely sequenced *Rhodobacterales* strains with a *fla1* flagellum (Fig. 2), the model organism *P. inhibens* DSM 17395 (clade 1) was the most suitable and therefore chosen for a systematic assessment of essential motility genes. *M. algicola* DG898 (clade 3B) and *C. sphaeroides* 2.4.1 (clade 8) both possess a second FGC of the *fla2*- and *fla3*-type, respectively, and were therefore not appropriate for a specific analysis of *fla1* genes (Fig. 1). *D. shibae* DSM 16493 (clade 5) was immotile on soft agar plates (20), which excluded this strain from a systematic screening campaign. *Epibacterium* sp. TM1040 (clade 1), which was isolated from the phycosphere of a dinoflagellate, has never been deposited in a public culture collection. However, this isolate in general represents a valuable reference organism, as the genes associated with motility have been intensively studied (16, 39, 40).

To identify genes essential for the swimming motility of *P. inhibens*, we generated a transposon insertion library and spotted individual mutants on swimming agar plates (17). Initially, 3,979 mutants were generated with insertion sites determined by sequencing (Table S3A). As this library covered only 1,767 (44.6%) of the *P. inhibens* genes, another 8,000 transposon mutants were analyzed, and the insertion sites were

determined only for those that showed impaired motility. In total, 158 non-motile transposon mutants were identified (Table S3B). For further analysis, we only considered genes within the FGC if they were hit, and additionally those genes outside the FGC for which at least two independent mutants were identified.

Within the FGC, 30 of the 36 genes were hit (Table 1; Fig. S3A). Multiple immotile transposon mutants were obtained for most of these genes, including those encoding the three conserved hypothetical proteins CP2, CP3, and CP4. The *flgD* mutant exhibited motility after prolonged periods of incubation. A mutant of the putative flagellar biosynthesis repressor *flbT* was obtained from the sequenced transposon library. However, in accordance with its presumed role, it was still motile. No mutants were generated for the genes *fliN* (PGA1_c35620), *fliL* (PGA1_c35650), *flgA* (PGA1_c35770), *flgB* (PGA1_c35830), *flgJ* (PGA1_c35890), and PGA1_c35740. Of note, an immotile *fliL* mutant was found in the motility screening of *Epibacterium* sp. TM1040 (39) and the gene might also be essential for the swimming motility of *P. inhibens*. In addition, transposon insertions in two other flagellar-associated genes located far upstream of the FGC in *P. inhibens*, namely *fliG* and CP1, resulted in immotile phenotypes. These genes were regarded as parts of the flagellar assembly machinery regardless of their upstream location, as they are found within the FGCs in other strains (CP1: Fig. 2A; *fliG*: Fig. 2B and C).

Outside the FGC, only eight genes were identified as essential for motility (Table 1). Among them were the three genes constituting the CtrA phosphorelay, *cckA*, *chpT*, and *ctrA*, which is in accordance with the study of Miller and Belas (16). In addition, non-motile mutants were identified for the putative regulatory protein *ntrX*, a homolog of the *flbD* gene of *C. vibrioides* (e-value = $5e^{-53}$) (41), as well as *chvG*, encoding a sensor protein, *rseP* (peptidase), and PGA1_c08510 (extracytosolic sigma factor). An insertion in *algC*, coding for the phosphomannomutase, linked motility to alginate biosynthesis. In the following, the focus was put on the roles that the four conserved hypothetical proteins CP1–4 might play in the assembly and function of the flagellum, as well as on the potential regulation by the CtrA phosphorelay.

## Roles of CP1–4 in flagella assembly of *P. inhibens* DSM 17395

The flagellum is built sequentially from the innermost to the outermost part, spanning two membranes from the cytosol into the extracellular space (Fig. 3A). In wild type *Salmonella enterica* serovar Typhimurium and static Δ*motA* or Δ*motB* mutants, proteins of the outermost part of the flagellum could be identified in culture supernatants using SDS-PAGE followed by amino acid sequencing (42). Here, we determined the exoproteome of *P. inhibens* wild type by mass spectrometry (Fig. 3B; Table S4). The flagellar protein secretion pattern consisted of P- and L-ring (FlgI and FlgH) proteins, the distal rod protein (FlgG), as well as the proteins of the hook (FlgE), hook-junction (FlgK), and hook-length control (FliK). By far, the strongest signal was found for flagellin (FliC). In the static transposon mutant strain Δ*motB*, the proximal rod protein (FlgF), the basal body protein (FliE), and a hook protein (FliK) were additionally detected. Thus, the proteins of functional and static flagella found outside of the cell overlapped between *P. inhibens* and *S. enterica*.

None of the four CPs were found in the exoproteome. Swimming motility could be restored by genetic complementation in all four mutants (Fig. 3C; Fig. S3B). To elucidate the role of these four conserved hypothetical proteins in the assembly of the flagellum, we compared the exoproteomes of the four CP mutants with those of the wild type and selected mutants (Fig. 3D). Within the ΔCP1 mutant, most of the proteins identified in the wild type and Δ*motB,* including the abundant filament protein FliC, were found. In ΔCP4, a reduced number of proteins were identified, among them FliC and the hook proteins FlgK and FlgE. The exoproteome of ΔCP2, ΔCP3, and Δ*fliI* contained only FlgH. No flagellar proteins were found for Δ*flgH* and the transposon mutant of the presumed master regulator of gene expression Δ*ctrA*.

**TABLE 1** Disrupted genes with impaired motility[a]

| Locus tag | Gene | Product | No. of mutants | Tn mutant | Insertion site |
|---|---|---|---|---|---|
| Flagella gene cluster (FGC) | | | | | |
| PGA1_c18800 | *fliG* | Flagellar motor switch protein FliG | 3 | 3290 | 1952268+ |
| PGA1_c23640 | CP1 | Conserved protein CP1 | 2 | 1440 | 2466771+ |
| PGA1_c35560 | *motB* | Chemotaxis protein MotB | 4 | 3361 | 3728713+ |
| PGA1_c35570 | *flgE* | Flagellar hook protein FlgE | 10 | 782 | 3729860+ |
| PGA1_c35580 | *flgK* | Flagellar hook protein FlgK | 3 | 4443 | 3731763+ |
| PGA1_c35590 | *flgL* | Flagellar hook protein FlgL | 3 | 2118 | 3732047+ |
| PGA1_c35600 | *flgI* | Flagellar P-ring protein FlgI | 6 | 3592 | 3733226− |
| PGA1_c35610 | *fliP* | Flagellar biosynthesis protein FliP | 1 | 4518 | 3735138+ |
| PGA1_c35630 | *fliH* | ABC transporter ATP-binding protein FliH | 1 | 3398 | 3734959− |
| PGA1_c35640 | *fliF* | Flagellar M-ring protein FliF | 8 | 3068 | 3737238+ |
| PGA1_c35660 | CP2 | Conserved protein CP2 | 3 | 1631 | 3738902− |
| PGA1_c35670 | CP3 | Conserved protein CP3 | 2 | 1034 | 3738994− |
| PGA1_c35680 | *motA* | Flagellar motor protein MotA | 1 | 5031 | 3740309+ |
| PGA1_c35690 | CP4 | Conserved protein CP4 | 6 | 2607 | 3742556+ |
| PGA1_c35700 | *sltF* | Lytic transglycosylase SltF | 1 | 4426 | 3743264− |
| PGA1_c35710 | *flhA* | Flagellar biosynthesis protein FlhA | 5 | 710 | 3745734− |
| PGA1_c35720 | *fliR* | Flagellar biosynthesis protein FliR | 3 | 4759 | 3746302+ |
| PGA1_c35730 | *flhB* | Flagellar biosynthesis protein FlhB | 2 | 4441 | 3747729− |
| PGA1_c35750 | *fliL* | Flagellar basal body-associated protein FliL | 4 | 2884 | 3748834− |
| PGA1_c35760 | *flgH* | Flagellar L-ring protein FlgH | 1 | 743 | 3749152− |
| PGA1_c35780 | *flgG* | Flagellar basal body rod protein FlgG | 6 | 3456 | 3750674− |
| PGA1_c35790 | *flgF* | Flagellar basal body rod protein FlgF | 2 | 2665 | 3751256− |
| PGA1_c35800 | *fliQ* | Flagellar biosynthesis protein FliQ | 2 | 4452 | 3751630+ |
| PGA1_c35810 | *fliE* | Flagellar hook-basal body protein FliE | 2 | 1897 | 3751893+ |
| PGA1_c35820 | *flgC* | Flagellar basal body rod protein FlgC | 1 | 4472 | 3752253+ |
| PGA1_c35840 | *fliI* | ATP synthase FliI | 1 | 1597 | 3753928− |
| PGA1_c35850 | *flbT* | Flagellar biosynthesis repressor FlbT | 1 | 2437 | 3755272+ |
| PGA1_c35860 | *flaF* | Flagellar biosynthesis regulatory protein FlaF | 4 | 3775 | 3755769+ |
| PGA1_c35870 | *fliC* | Flagellin FliC | 2 | 4458 | 3756862− |
| PGA1_c35880 | *flgN* | FlgN-like protein | 3 | 3876 | 3757350− |
| PGA1_c35900 | *fliK* | Flagellar hook-length control protein FliK | 3 | 3948 | 3758574− |
| PGA1_c35910 | *flgD* | Flagellar basal body rod modification protein FlgD | 1 | 4473 | 3759992+ |
| Other | | | | | |
| PGA1_c01160 | *chvG* | Sensor protein ChvG | 6 | 4974 | 110670+ |
| PGA1_c14260 | *rseP* | Peptidase M50 | 3 | 4436 | 1481491+ |
| PGA1_c14360 | *ctrA* | Cell cycle response regulator CtrA | 3 | 4407 | 1490614− |
| PGA1_c14810 | *ntrX* | Nitrogen assimilation regulatory protein NtrX | 11 | 4516 | 1536811+ |
| PGA1_c15240 | *cckA* | Two-component system sensory histidine kinase CckA | 5 | 4437 | 1588155+ |
| PGA1_c15530 | *chpT* | Phosphotransfer protein ChpT | 3 | 4380 | 1616227− |
| PGA1_c24340 | *algC* | Phosphomannomutase AlgC | 3 | 4548 | 2539880− |
| PGA1_c08510 | *ecf* | RNA polymerase sigma factor ECF | 2 | 4491 | 879262+ |

[a]The identifier of one representative mutant (Table S3) and its insertion site and direction are provided.

Next, we used electron microscopy to test whether, in particular, ΔCP1 was still able to assemble a flagellum. Only a few flagellated cells and additionally some broken flagella were found in the wild type (Fig. S4A). No flagella were detected in the transposon mutant of *ctrA*, which served as a negative control. Furthermore, no flagella were present in the transposon mutants of the flagellar L-ring protein FlgH and the export ATPase protein FliI (Fig. S4B). Surprisingly, no flagella could be detected in CP1 mutant cells, although the protein secretion pattern largely resembled that of the wild type. In some ΔCP1 cells, an unusual, yet undefined structure was detected at one cell pole instead of a

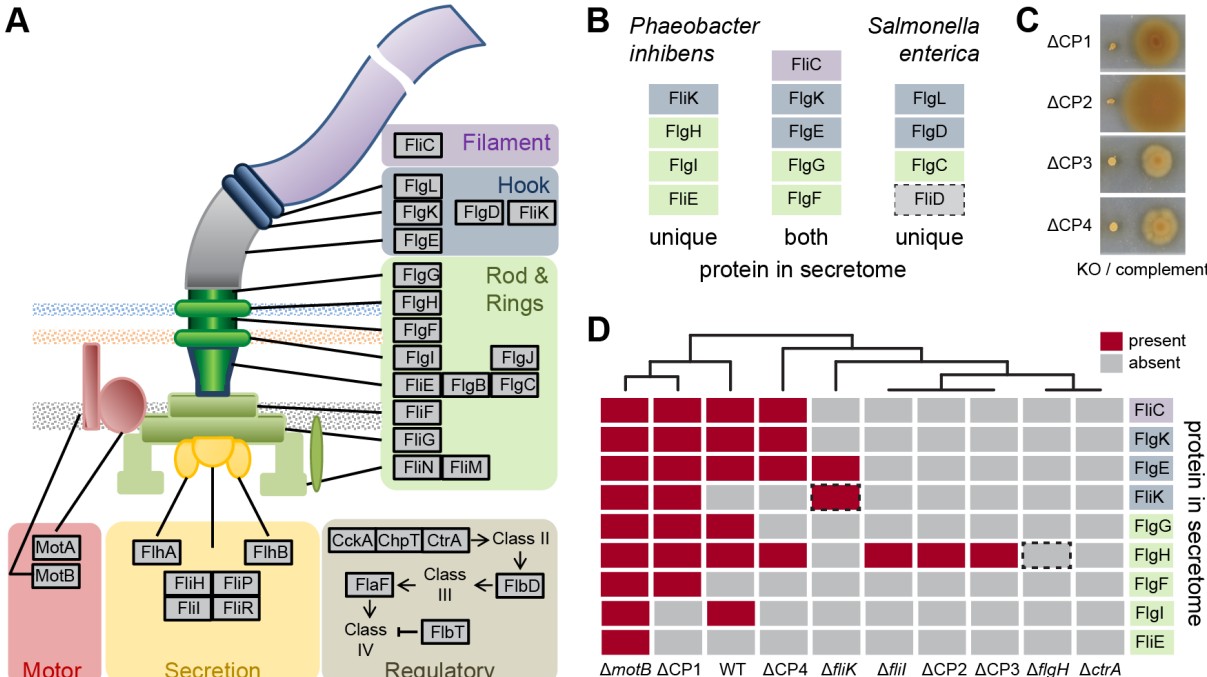

**FIG 3** Composition of the *P. inhibens* flagellum and comparative exoproteome analysis. (A) Schematic illustration of a bacterial flagellum with the spatial position and function of the proteins highlighted. (B) Comparison of the exoproteome of *P. inhibens* (wild type and Δ*motB*; current study) *and S. enterica* (42) showing proteins identified in one or both species. Coloring according to the function. FliD, absent in *P. inhibens,* is marked by a dotted square. (C) Motility assay of CP1-4 transposon knock-out (KO) mutants and the respective genetic complements. (D) The presence (red) and absence (gray) of flagellar proteins in the exoproteome of *P. inhibens* wild type and transposon mutants for CP1–4 as well as selected flagella genes and *ctrA* as controls. Dotted squares mark proteins of knocked-out genes in the respective strains.

flagellum (Fig. S4C). We were not able to detect flagella in the mutant strains of CP2 and CP3 using electron microscopy.

## Regulatory control of gene expression in *P. inhibens* by the CtrA phosphorelay

We complemented the three *P. inhibens* mutants (Δ*ctrA*, Δ*cckA,* and Δ*chpT*) coding for the CtrA phosphorelay genes by expressing functional copies on a plasmid (Fig. 4A; Fig. S3C). We then compared the transcriptomes of the three mutants with the *P. inhibens* wild type. All three knockouts resulted predominantly in a downregulation of 127 shared genes, confirming the role of *ctrA* as a transcriptional activator (Fig. 4B; Table S5). A large overlap of 70 additional regulated genes was found between the *ctrA* and *cckA* mutants, while the overlap with *chpT* was smaller. Significant changes in expression were restricted to the chromosome as well as the 262 kb and 78 kb chromids (Fig. 4C). Of note, the strongest repression was seen at the chromosomal origin (*oriC*) and terminus (*terC*) of replication. The latter region consisted of a 35.6 kb prophage with 48 genes. The flagellar gene cluster was located near *oriC,* and three orphan flagellar genes were silenced to an equal extent in all mutants of the CtrA phosphorelay (Fig. 4D). The strongest response was seen for the flagellin-encoding *fliC* gene, whose expression was reduced 64-fold. The only exception was the orphan second copy of the flagella motor protein *motA*, which was more than 12-fold upregulated in Δ*ctrA* and Δ*cckA*, and 6-fold downregulated in Δ*chpT*.

Three signaling pathways were affected by knockouts of the CtrA phosphorelay: (i) The *P. inhibens* chemotaxis control consists of a complete set of eight chemotaxis (*che*) signal transduction genes encoded on the 262 kb chromid, which were all downregulated up to 32-fold in all three mutants (Fig. 4E). A chromosomal gene coding for a chimeric

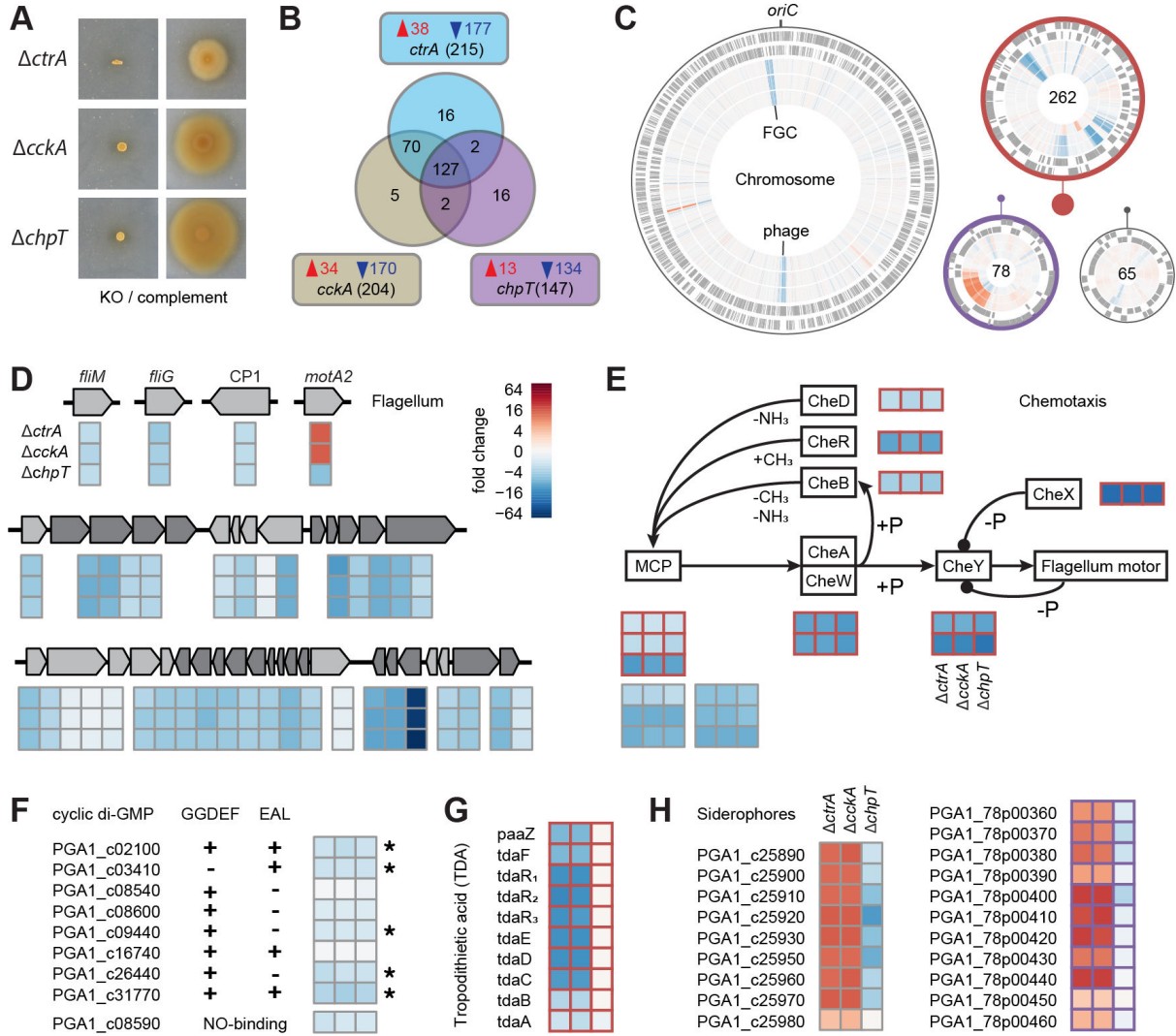

**FIG 4** The CtrA phosphorelay regulon of *P. inhibens*. (A) Motility assay of phosphorelay transposon mutants and the respective genetic complements. (B) Unique and overlapping genes with a significantly different expression between the three mutant strains and the wild type. Upregulated and downregulated genes are shown in red and blue, respectively. (C) Differential gene expression on the chromosome and three chromids. From outer to inner ring, genes on the plus and minus strand, and the fold change between Δ*ctrA*, Δ*cckA*, and Δ*chpT* and the wild type are shown. The size of the plasmids in relation to the chromosomes is indicated by closed circles. Heatmaps presenting differential expression of (D) flagella, (E) chemotaxis, (F) cyclic di-GMP turnover, (G) antibiotic biosynthesis, and (H) siderophore genes highlight similarities and differences between the three knockout strains. The four genes outside the flagella gene cluster (FGC) are named operons within the FGC (PGA1_c35560 to PGA1_c35910) and are shown in two different shades of gray. Significantly differential expression of cyclic di-GMP turnover genes is indicated by an asterisk. The TDA gene cluster ranges from PGA1_262p00800 to PGA1_262p00980. Genes on the 262 kb and 78 kb chromids are framed in orange and purple, respectively.

protein containing CheB and CheR domains did not change in expression. Attractants are sensed by 12 genes encoding methyl-accepting chemotaxis proteins (MCPs). Three MCPs encoded on the 262 kb chromid and six of the nine MCPs on the chromosome were downregulated. (ii) Another signaling molecule that is important for the switch between motile and sessile states is cyclic-di-GMP (43). It is synthesized by diguanylate cyclases (GGDEF domain) and hydrolyzed by phosphodiesterases (EAL domain), of which *P. inhibens* contains four and one, respectively (Fig. 4F). In addition, three genes were found that encode proteins with both domains. Five of these genes were repressed between twofold and fourfold in the three mutants. (iii) The *P. inhibens* quorum-sensing system consists of two autoinducer synthase genes. One is encoded in an operon with the corresponding LuxR-type transcriptional regulator (*pgaRI₁*). Neither gene was

differentially expressed. The second synthase, termed *pgaI₂*, is located downstream of a hybrid histidine kinase/response regulator. This gene was downregulated approximately fourfold in all three mutants (see below).

Two traits were exclusively regulated in the *ctrA* and *cckA* mutants. The gene cluster for the biosynthesis of the antibiotic tropodithietic acid (TDA) was eightfold downregulated in these strains (Fig. 4G). By contrast, one chromosomal and one chromid-encoded cluster for siderophore production and uptake were eightfold upregulated only in Δ*ctrA* and Δ*cckA*, similar to the *motA2* gene (Fig. 4H). The gene transfer agent (GTA) gene cluster, a conserved target of CtrA (44, 45), was not expressed in any of the mutants.

## Comparison of the CtrA regulon of *P. inhibens* and *D. shibae*

We next compared the new *P. inhibens* data set with a previous transcriptome analysis of *D. shibae* wild type and three CtrA phosphorelay mutants (46). Physiological and technical differences have to be considered here, as the samples originated from different growth media (MB *versus* artificial seawater), and gene expression in *D. shibae* has been monitored using microarrays. Nevertheless, the identification of a core regulon was possible. Both *Roseobacteraceae* share 2,333 orthologs. Of these genes, 146 were expressed differentially in either one or both species compared to the respective wild type. Using the same cut-off values (absolute fold change >2 and FDR < 0.05) for both data sets revealed only about 40 commonly downregulated genes in the respective mutants of both strains (Fig. 5A). Of these, 31 were located in the FGCs, including the newly described genes encoding CP1–4 (Fig. 5B). Several genes fell only slightly below the cut-off value (Table S6), which indicates further commonly regulated traits.

Similarities and differences were found for the shared regulatory genes (Fig. 5C). In both species, knocking out one of the phosphorelay genes did not affect the expression of the other two. A remarkable difference was found in the quorum-sensing system. Both strains have two chromosomally encoded autoinducer synthases. However, the orthologs showed inverse regulation. The *pgaIR₁* and *pgaI₂* homologs were only downregulated in *D. shibae* and *P. inhibens*, respectively, and did not change their expression in the other species. In contrast to *P. inhibens*, the *pgaI₂* homolog of *D. shibae* is located in an operon with the corresponding transcription factor. The *D. shibae* cyclic-di-GMP turnover system consists only of two diguanylate cyclases (DGCs) and two phosphodiesterases (PDEs), all with homologs in *P. inhibens*. Both *pde* genes, but only one *dgc* gene and the associated *hnox* gene, were downregulated in both roseobacter strains. The extended CtrA regulon of *D. shibae* contains more genes with sensory or regulatory function. Three histidine kinases, two of them homologs of *divL* from *C. crescentus* (47), showed a strong repression in all *D. shibae* mutants, but only *divL1* was repressed in *P. inhibens* Δ*ctrA* and Δ*cckA*. The transcriptional activator *gafA* and an anti-sigma-factor system *rbsVW* are involved in the control of GTA production (48, 49). The *P. inhibens* homologs were not repressed in the phosphorelay mutants. This is in accordance with the inactivity of the GTA gene cluster. In the *D. shibae* mutants, these genes were downregulated along with GTA and competence genes (Fig. 5C and D).

A tight adherence (TAD) pilus gene cluster associated with roseobacter dendritic motility (20) was exclusively repressed in *D. shibae* (Fig. 5D). By contrast, the nitric oxide (NO) reduction operon was repressed in *P. inhibens*, while the homologs in *D. shibae*, which are part of a complete denitrification pathway (50), were not affected by the CtrA phosphorelay knockouts. Finally, *dprA*, encoding a DNA repair protein, was downregulated in the mutants of both species, but was much more pronounced in *P. inhibens*. Three other regulators of the stress response were slightly downregulated in at least one *D. shibae* mutant, but their expression did not change in the *P. inhibens* mutants.

## DISCUSSION

A phylogenetically broad taxon sampling with more than 300 genomes allowed us to draw reliable conclusions about the origin, evolution, and current distribution of three different flagellar systems in *Rhodobacterales*. The one order of magnitude larger data set

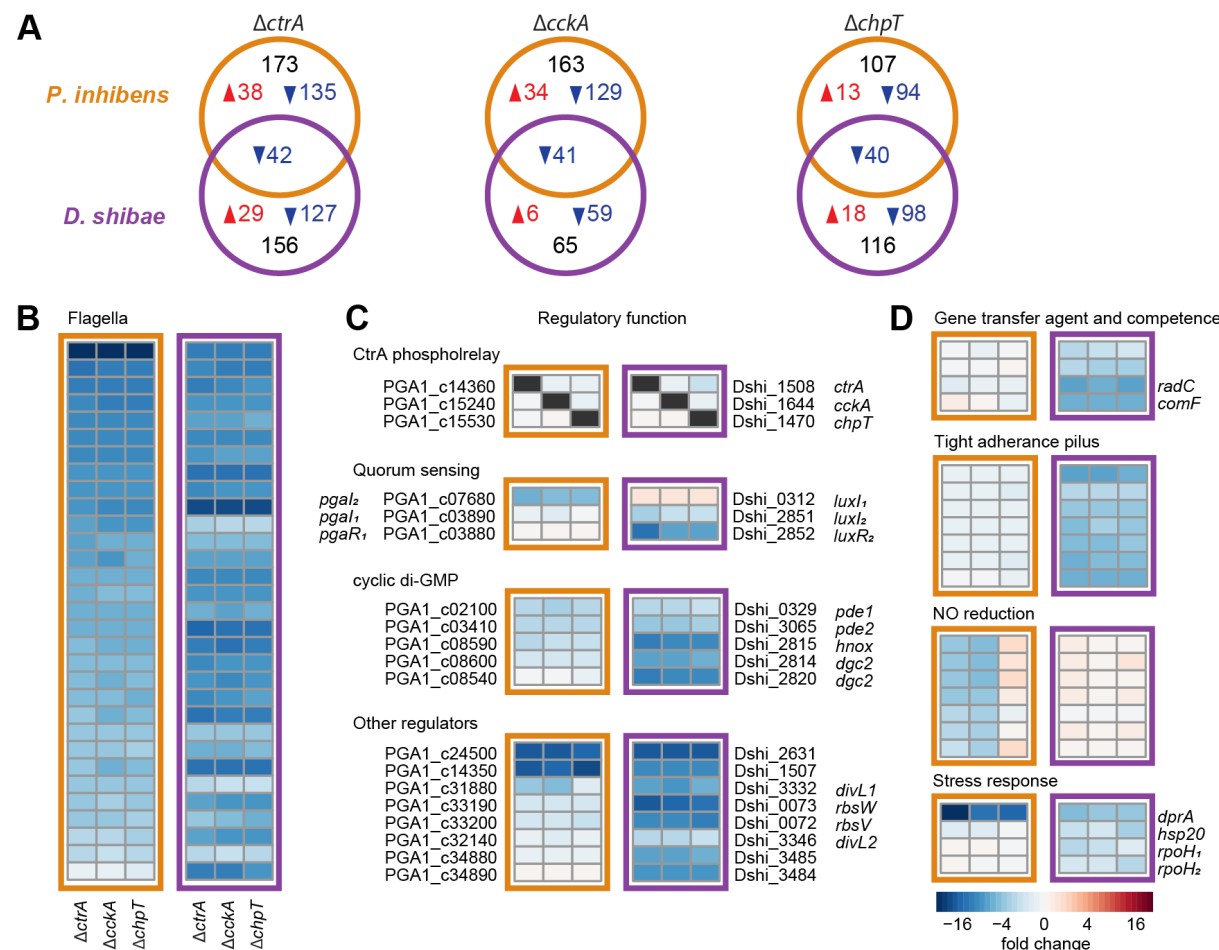

**FIG 5** Comparison of the *P. inhibens* and *D. shibae* CtrA phosphorelay regulon. (A) Unique and overlapping orthologous genes with significantly different expression between the three mutant strains and the wild type of *P. inhibens* and *D. shibae*. Upregulated and downregulated genes are shown in red and blue, respectively. (B) Flagellar genes, (C) regulatory genes, and (D) other CtrA targets as side-by-side heatmap comparison of *P. inhibens* (left) and *D. shibae* (right) marked in orange and purple, respectively.

corroborated the conclusions from the previous census conducted a decade ago (30), namely (i) the presence of three different *Rhodobacterales*-specific flagellar systems (*fla1*, *fla2*, and *fla3*), (ii) the predominance of the *fla1*-type flagellum, and (iii) the co-occurrence of a maximum of two FGCs (*fla1/fla2* or *fla1/fla3*) per genome. The functionality of two different flagellar systems in one bacterium has been exemplified for *C. sphaeroides* (51), and the benefit of an additional FGC is likely related to a more flexible regulation of swimming motility (see below). The absence of the combination *fla2/fla3* is consistent with the main finding of the current study that *fla1* represents the archetypal flagellum in the order *Rhodobacterales*.

All three flagellar types are present in *Roseobacteraceae* and *Paracoccaceae*, but *fla1* was also found in the four basally branching genera *Meinhardsimonia*, *Albimonas*, *Oceanicella,* and *Rubrimonas*, which represent taxa of additional but as yet undescribed families within the order *Rhodobacterales*. In particular, the presence of *fla1* in *Rubrimonas cliftonensis* is noteworthy, as this species represents the earliest branching lineage of *Rhodobacterales* with a FGC (1). Due to the absence of *fla1* in other alphaproteobacterial orders and in light of its vertical evolution, the observed distribution suggests that this characteristic flagellar system originated in the common ancestor of *Rhodobacterales*.

The synchronicity of *fla1* and organismic evolution reflects the importance of swimming motility for *Rhodobacterales*. The vertical inheritance of the most abundant flagellar type *fla1* is also supported by its exclusively chromosomal localization. By

contrast, the scattered distribution of *fla2* and *fla3* in the phylogenomic tree and their taxonomically inconsistent flagellar phylogenies provided clear evidence that these FGCs were subject to frequent HOTs. The promiscuity, especially of *fla2,* is reflected by an extrachromosomal localization of FGCs on replicons of 10 different compatibility groups representing the four most common plasmid types of *Rhodobacterales* (DnaA-like, RepA, RepB, and RepABC). This finding is analogous to the extrachromosomal localization of photosynthesis gene clusters (PGCs) within the *Rhodobacterales* and their lateral distribution via HOTs (52–54). The clustering of functionally connected genes (FGC and PGC) provided the basis for the horizontal exchange of these complex bacterial traits. This reflects the crucial role of horizontal gene transfer in alphaproteobacterial evolution beyond rapid adaptations to heavy metal or antibiotic pollution in the environment, due to the transfer of single or a few genes (55, 56).

Our systematic analysis of genes that are essential for swimming motility in *P. inhibens* indicated that the horizontal exchange of an FGC across genus borders, leading to a functional flagellum, is very challenging. All structural genes and regulators must be exchanged *en bloc,* and the components of the complex nanomachine have to be properly expressed after the genetic "transplantation." Accordingly, only two examples of ancient horizontal recruitment of xenologous FGCs from other alphaproteobacterial orders have been reported for *Rhodobacterales*, namely the original acquisition of *fla2* and *fla3*. In addition to the recent exchange of *fla2*-type FGCs within the *Rhodobacterales*, an alternative explanation for the frequent localization of *fla2* on plasmids could be the necessity of spatial linkage of the genes to the cell pole, analogous to the positioning of *fla1* of *P. inhibens* close to the *oriC*. A coupled transcription, translation, and membrane insertion ("transertion") of the bacterial flagellum could thus result in a (sub-)polar localization in the cell (30, 57).

The archetypal *fla1* gene cluster of *Rhodobacterales* has probably acquired the four new genes (CP1–4) after the split from the other alphaproteobacterial orders. Although their functions remain enigmatic, the corresponding proteins are essential for the formation of a functional flagellum. CP1 and CP3 might act comparably late in the assembly process, as rod, hook, and filament proteins could still be detected outside the cell. Possible roles as chaperones, or in prevention of premature assembly, like for the recently described FlhE of *S. enterica* (58), should therefore be considered. Only a very limited number of essential motility genes were located outside the FGC. In general, we hit genes more than once, which indicates that the coverage of our screening was sufficient for a comprehensive identification of motility factors. The regulators identified, in addition to the CtrA phosphorelay system, might play a role in tuning FGC expression in response to specific environmental signals or affect its timing. The influence of the phosphomannomutase AlgC is of particular interest, as we previously found biosynthesis of rhamnose, sharing a precursor with alginate, to be essential for motility and biofilm formation (17, 19). Notably, in this case, only the loss of the whole *rmlCBDA* rhamnose operon, but not single gene knockouts, led to a completely immotile phenotype. This explains why we did not hit these genes in our transposon mutant screening. In this regard, we also have to consider that motility genes acting only in concert may have been overlooked.

We found that no quorum-sensing genes are essential for motility. This is in accordance with the upregulation of flagella genes and increased motility previously observed in *P. inhibens* QS knock-outs (59). In *D. shibae*, knockout of autoinducer biosynthesis downregulates flagella gene expression (31, 60). The physiological consequences remain open, as the strain is immotile in the standard soft agar assay (20). However, the CtrA core regulon of motility and cyclic-di-GMP turnover and its integration into the QS system are conserved in these two phylogenetically distant species of *Roseobacteraceae*. It is likely central for the widespread swim-and-stick lifestyle in this family of marine bacteria (19, 20) and also essential for interactions with algal hosts (16, 18). Nonetheless, there are differences between the two strains in terms of the CtrA-mediated expression of orthologous genes. Some of them can be *bona fide* attributed to differences in

cultivation conditions or the timing of sampling. Expression of the GTA gene cluster, the respective activator *gafA* (49), and modulators *rbsWV* (48) was almost absent in *P. inhibens*. This contrasts with the weak but notable expression in *D. shibae* (31, 46). These two regulatory systems were first unraveled in *Rhodobacter capsulatus*, where they are also controlled by CtrA and QS (61). They are probably universally conserved in *Rhodobacterales* and should, under the right environmental conditions, also be functional in *P. inhibens*. However, it is also possible that the expression of the GTA gene cluster is inhibited by the prophage located at the terminus.

Differences in the regulation of the denitrification genes between *P. inhibens* and *D. shibae* might have evolved as an adaptation to specific environmental niches. *P. inhibens*, which has only an incomplete denitrification pathway, uses its chromid-encoded nitrite reductase (*nirKV*) under oxic conditions to reduce nitrite released by the coccolitho-phore *Gephyrocapsa huxleyi* (formerly *Emiliania huxleyi*) to NO, which, in turn, promotes programmed cell death of the alga (15). By contrast, the presence of a complete chromosomal denitrification pathway allows *D. shibae* to grow anaerobically using $NO_3^-$ as a terminal electron acceptor (50). Here, the anaerobic activators of the denitrification genes, in turn, repress the CtrA phosphorelay, while inverse regulation was not observed (62). Thus, oxygen availability could restrict expression to either denitrification or flagella, contrasting their co-expression in *P. inhibens*.

The control of motility has been suggested to be the ancient function of the alphaproteobacterial CtrA phosphorelay (63). A finding that requires further attention is the co-localization of CtrA phosphorelay genes with two of the scattered flagellar operons in *Rh. rubrum*. The *Rhodospirillales* are one of the earliest branching alphapro-teobacterial phyla, and it is tempting to speculate that the chromosomal separation of the phosphorelay genes from the flagella genes reflects the evolution of CtrA toward a more generalized and finally essential cell cycle regulator (24). Likewise, the fixing of the phosphorelay and regulon genes at different chromosomal positions might have played a role in this process (64). In *P. inhibens* and *D. shibae*, the flagella genes are located at the origin of replication (*oriC*), while *ctrA* and *cckA* are located at the terminus; thus, the distance between regulator and target is maximized.

This study was focused on an important roseobacter model organism that possesses only the main *fla1*-type flagellum. The widespread but scattered occurrence of dual flagellar systems indicates a role in niche adaptation at the genus level. There is also the question of how the expression of two FGCs is coordinated, and so far, we only have first indications from *C. sphaeroides*. Note that in this bacterium, the *fla1* and *fla3*-type FGC were originally named in order of their discovery, fla1 and fla2, respectively. Both *fla1* and *fla3*-type result in the formation of a single polar flagellum. The *fla1*-type flagellum is co-activated with gas vesicle formation by CtrA (65, 66). Both traits together allow the bacterium to roam the oxygenated water surface. On the other hand, the expression of the *fla2*-type FGC is stimulated together with the PGC by oxygen depletion (51). Under these conditions, light can provide energy for movement. *Rhodobacterales* species with distinct flagella types were isolated from a variety of habitats. Future studies might reveal different regulatory mechanisms and help to better understand the evolutionary benefits of a bacterial chassis with two propeller systems.

## MATERIALS AND METHODS

### Phylogenomic analysis

For multilocus sequence analyses (MLSA), the core genome of 306 *Rhodobacterales* genomes deposited at the NCBI database was determined with Proteinortho5 (67) using the parameters "-selfblast -singles -cov = 60 e =1 e − 08 -identity = 25." Core genes that were not duplicated in any of the genomes analyzed were aligned using MUSCLE (68). Partially aligned C- and N-terminal trailing regions were clipped. Poorly aligned internal alignment regions were filtered out using Gblocks (69). To reduce

the influence of potential horizontal gene transfer, each orthologous group was then preliminarily clustered using fasttree, and the mean and standard deviation of summed branch lengths were calculated for all trees. Markers that resulted in tree topologies with summed branch lengths greater than the calculated mean plus two standard deviations were dismissed. Only 36 of 306 analyzed genomes were complete, and seven draft genomes contained more than 200 contigs. To counteract missing genes due to incomplete genome assembly, sets of core genes present in all genomes (0 missing genes), and sets with 1, 3, and 15 missing genes were analyzed. The most stringent data set without missing genes resulted in an alignment with 44,988 amino acid positions (100% criterion: 168 marker genes), while 1 missing gene corresponded to 85,058 aa (99.8%: 295 genes), 3 missing genes to 123,793 aa (99%: 417 genes), and 15 missing genes to 173,174 aa (95%: 590 marker genes). For the current study, we chose the data set with three missing genes (99% criterion), resulting in a selection of 417 marker genes with 123,793 conserved amino acid residues. The concatenated alignment was clustered using the maximum likelihood method implemented by RaxML (70) with the "rapid bootstrap" method and 250 bootstrap iterations. Additional node support was determined using IQ-TREE (71) with the option of 1,000 rounds of ultrafast bootstrap approximation (72). Phylogenetic trees with associated metadata were visualized with the R package ggtree (73).

## Analysis of flagellar gene clusters

The distribution of flagellar systems in the 306 *Rhodobacterales* genomes was investigated by comprehensive BLASTP searches using the three previously characterized FGCs from *P. inhibens* DSM 17395 (*fla1*; CP002976.1: PGA1_c35560 to PGA1_c35910), *M. algicola* DG898 (*fla2*; CP010860.1: MALG_04521 to MALG_04566), and *C. sphaeroides* 2.4.1 (*fla3*; CP000143.2: RSP_0032 to RSP_0083) as references (30). The abundance of 270 whole-genome shotgun (WGS) genomes and only 36 complete genomes in the data set of 306 investigated *Rhodobacterales* hindered the unambiguous identification of extrachromosomal FGCs. Therefore, the presence of plasmid replication proteins (DnaA-like, RepA, RepB, or RepABC) on FGC-containing contigs was used to identify plasmid-/chromid-encoded flagellar systems. The corresponding compatibility groups were determined based on our established phylogenetic classification scheme (35, 74). All other FGCs were regarded as chromosome-encoded.

## Cultivation conditions

*P. inhibens* was routinely grown in marine broth (MB) medium (Carl Roth GmbH, Karlsruhe, Germany) under constant shaking at 28°C. Whenever grown under antibiotic pressure (120 µg/mL kanamycin or 40 µg/mL gentamicin, Carl Roth GmbH), half-strength MB medium was used for cultivation to prevent reduction of antibiotic activity. The generated transposon mutant strains and complementation strains are available from the authors upon request.

## Generation and identification of transposon mutants

*P. inhibens* DSM 17,395 cells were electroporated for uptake of the Transposon EZ-Tn5 <KAN-2> (Cambio, Cambridge, UK). In total, 4,954 kanamycin-resistant colonies were picked and individually stored as glycerol stocks. The insertion site of 3,979 mutants could be identified by sequencing the product of two consecutive PCRs. The product of the first PCR using an arbitrary and a specific primer was amplified using two specific primers. Used primers are listed in Table S7. For a second round of screening, an additional 8,000 Km-resistant mutants were picked, and the insertion site was only determined in the case of impaired mobility.

## Generation of complementation strains

The single CP1 gene and the whole CP4-containing operon were cloned together with the upstream intergenic region into the broad-host-range expression vector pBBR1-MCS5. For CP2 and CP3, the constitutively active gentamicin promoter was cloned in front of the ORF. Used primers are listed in Table S7. Precultures for experiments were grown under selection pressure to prevent plasmid loss, while main cultures were grown without selection pressure. *P. inhibens* DSM 17395 carrying the empty vector was used as a control in the respective experiments.

## Motility assay on soft agar

The flagellar motility of *P. inhibens* DSM 17395 and its mutants was investigated on 0.3% (w/v) Marine Broth (MB) soft agar plates. A preculture of each strain was inoculated in 4.5 mL of the respective liquid medium and incubated at 28°C in test tubes under moderate shaking. Test plates were inoculated by spotting 3 µL of the pre-culture on the soft agar and incubated at 28°C for 2–9 days.

## Exoproteome analysis

Extracellular proteins were prepared from filtered (0.22 µm) culture supernatants of mid-exponential cultures as described before (75). In brief, proteins were precipitated with trichloroacetic acid overnight at 4°C, collected by centrifugation, and washed with ethanol prior to resuspension in urea/thiourea buffer (7 M urea, 2 M thiourea, 30 mM Tris, pH 8.5) and storage at −80°C. Fifty microliter of a 1 µg/µL solution of prepared extracellular protein (in urea buffer: 8 M urea, 0.4 M $NH_4CO_3$) was subjected to reduction at 55°C for 30 min using dithiothreitol (45 mM) and subsequent alkylation by iodoacetamide (100 mM; 15 min in the dark). The solution was diluted with water to a concentration of 2 M urea prior to overnight tryptic digest (Trypsin Gold, Promega, Mannheim, Germany) at 37°C.

Peptides (corresponding to 1 µg protein) were separated by nanoLC (Ultimate 3000 nanoRSLC, ThermoFisher Scientific, Germering, Germany) equipped with a 25 cm analytical column (C18, 2 µm bead size, 75 µm inner diameter; ThermoFisher Scientific) operated in trap-column mode (C18, 5 µm bead size, 2 cm length, 75 µm inner diameter; ThermoFisher Scientific) using a 180 min linear gradient as previously described (76). The eluent was continuously analyzed by an online-coupled ion-trap mass spectrometer (amaZon speed ETD; Bruker Daltonik GmbH, Bremen, Germany) using a captive spray ion source (Bruker Daltonik GmbH). Protein identification was performed via the ProteinScape platform (version 3.1; Bruker Daltonik GmbH) on an in-house Mascot server (version 2.3 Matrix Science Ltd; London, UK) against the translated genome of *P. inhibens* DSM 17395 and applying a target-decoy strategy with the following settings: enzyme, trypsin; single missed cleavage allowed; modifications, carbamidomethylation (C) fixed and oxidation (M) variable; Peptide mass tolerance 0.3 Da; monoisotopic; MS/MS mass tolerance 0.4 Da; significance threshold $P < 0.05$; false discovery rate <1.0%. Only proteins identified in at least two replicates were considered.

## Electron microscopy

Samples were fixed with 2% glutaraldehyde and washed twice in TE buffer (10 mM Tris, 2 mM EDTA, pH 6.9), dehydrated with a graded series of acetone (10%, 30%, 50%, 70%, 90%, and 100%) for 15 min on ice and further dehydrated in 100% acetone at room temperature. After critical-point drying with liquid $CO_2$ (CPD300, Leica), samples were sputter-coated with palladium–gold and imaged in a Zeiss Merlin field emission scanning electron microscope (Zeiss AG, Oberkochen, Germany) with an acceleration voltage of 5 kV. Images were taken with the Zeiss Smart SEM software version 5.04.

## Transcriptomics

Total RNA was isolated with the RNeasy kit (Qiagen, Hilden, Germany) according to the manufacturer's protocol and including the optional DNase digestion step. Depletion of rRNA was performed with the mRNA Ribo-Zero magnetic kit (Epicentre, Madison, WI, USA) according to the manufacturer's instructions. The library was prepared from ribosomally depleted total RNA using the Scriptseq v2 RNA-seq (RNA sequencing) library preparation kit (Epicentre) according to the manufacturer's protocol. The libraries were sequenced on Illumina HiSeq2500 (Illumina, San Diego, CA, USA) in a 50-cycle, single-ended run. The demultiplexed raw fastq files were quality controlled using the FASTQ-mcf suite (https://github.com/ExpressionAnalysis/ea-utils). Low-quality bases (phred-score <30) and identified Illumina adaptors were clipped from the sequences. Reads were mapped to reference genomes using bowtie2 (77) in end-to-end mode with default parameters. A table with reads per protein-coding sequence was generated from each indexed bam file using featureCounts (78) with default parameters.

The R package EdgeR was used for assessing differential gene expression (79). Dispersion was estimated in robust mode. Genewise negative binomial generalized linear models were fitted (glmFit), and significance was calculated with likelihood ratio tests (glmLRT). Only genes with an absolute log2 fold change >1 and a false discovery rate <0.05 were considered significantly differentially expressed. For comparative transcriptomics, orthologs between *P. inhibens* and *D. shibae* were determined using the bidirectional best BLAST hit approach with an e-value cutoff = $10^{-10}$. Functional groups were assigned manually. From the published *D. shibae* microarray data comparing the CtrA phosphorelay mutants, only the data set recorded in the exponential growth phase was considered. The core regulon was determined with a significance cut-off as above. In addition, genes were considered when the expression of orthologs changed in the same direction in the corresponding mutants of both species.

## ACKNOWLEDGMENTS

We thank the reviewer for their critical and constructive assessment of the manuscript.

This project was funded by the German Research Foundation (DFG) through the Transregio SFB TRR-52 Roseobacter.

## AUTHOR AFFILIATIONS

[1]Laboratory of Anoxygenic Phototrophs, Institute of Microbiology of the Czech Academy of Science – Centre Algatech, Třeboň, Czech Republic
[2]Leibniz Institute DSMZ – German Collection of Microorganisms and Cell Cultures, Braunschweig, Lower Saxony, Germany
[3]Institute for Biological Interfaces 5, Karlsruhe Institute of Technology, Karlsruhe, Baden-Württemberg, Germany
[4]General and Molecular Microbiology, Institute for Chemistry and Biology of the Marine Environment (ICBM), Carl von Ossietzky University of Oldenburg, Oldenburg, Lower Saxony, Germany
[5]Genome Analytics, Helmholtz Centre for Infection Research, Braunschweig, Lower Saxony, Germany
[6]Institute of Microbiology, Technical University of Braunschweig, Braunschweig, Lower Saxony, Germany

## AUTHOR ORCIDs

Jürgen Tomasch ● http://orcid.org/0000-0002-3914-2781
Ralf Rabus ● http://orcid.org/0000-0001-5536-431X
Jörn Petersen ● http://orcid.org/0000-0001-6223-5575

## FUNDING

| Funder | Grant(s) | Author(s) |
|---|---|---|
| Deutsche Forschungsgemeinschaft | SFB TRR-52 | Ralf Rabus |
| | | Jörn Petersen |

## AUTHOR CONTRIBUTIONS

Jürgen Tomasch, Conceptualization, Formal analysis, Visualization, Writing – original draft | Pascal Bartling, Conceptualization, Formal analysis, Investigation, Writing – original draft | John Vollmers, Formal analysis | Lars Wöhlbrand, Formal analysis, Investigation, Writing – review and editing | Michael Jarek, Data curation | Manfred Rohde, Visualization | Henner Brinkmann, Formal analysis | Heike M. Freese, Formal analysis, Visualization, Writing – review and editing | Ralf Rabus, Conceptualization, Funding acquisition, Writing – review and editing | Jörn Petersen, Conceptualization, Formal analysis, Funding acquisition, Project administration, Visualization, Writing – original draft

## DATA AVAILABILITY

The mass spectrometry proteomics data have been deposited in the ProteomeXchange Consortium (https://proteomecentral.proteomexchange.org) via the PRIDE partner repository (80) with the data set identifier PXD061497. RNAseq data have been deposited in the NCBI Gene Expression Omnibus Database (https://www.ncbi.nlm.nih.gov/geo/) with the identifier GSE291569.

## ADDITIONAL FILES

The following material is available online.

### Supplemental Material

**Figure S1 (mSystems00419-25-s0001.pdf).** Phylogenomic trees of *Rhodobacterales* and their flagella.
**Figure S2 (mSystems00419-25-s0002.pdf).** Subtrees for *fla2* and *fla3*-type flagella.
**Figures S3 and S4 (mSystems00419-25-s0003.pdf).** Motility assays and electron microscopy of *P. inhibens* strains.
**Supplemental Tables (mSystems00419-25-s0004.xlsx).** Tables S1-S7.

### Open Peer Review

**PEER REVIEW HISTORY (review-history.pdf).** An accounting of the reviewer comments and feedback.

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
