## [Reviewer comments · mSystems]

Structural and regulatory determinants of flagellar motility in Rhodobacterales – The archetypal flagellum of *Phaeobacter inhibens* DSM 17395

Jürgen Tomasch, Pascal Bartling, John Vollmers, Lars Wöhlbrand, Michael Jarek, Manfred Rohde, Henner Brinkmann, Heike Freese, Ralf Rabus, and Jörn Petersen

Corresponding Author(s): Jürgen Tomasch, Institute of Microbiology, CAS Centre Algatech

Review Timeline:

Submission Date:	March 22, 2025
Editorial Decision:	May 4, 2025
Revision Received:	May 18, 2025
Accepted:	June 2, 2025

Editor: Maria Scrase

Reviewer(s): Disclosure of reviewer identity is with reference to reviewer comments included in decision letter(s). The following individuals involved in review of your submission have agreed to reveal their identity: Elias Javier Mongiardini (Reviewer #1)

Transaction Report:

DOI: <https://doi.org/10.1128/msystems.00419-25>

Re: mSystems00419-25 (Structural and regulatory determinants of flagellar motility in Rhodobacterales - The archetypal flagellum of Phaeobacter inhibens DSM 17395)

Dear Dr. Jürgen Tomasch:

Before to reach a final decision, I ask you to address all the points raised by the reviewer.

Revision Guidelines

Sincerely,
Maria Scarscia
Editor
mSystems

MS mSystems00419-25 revision:

This MS focuses on flagellar motility in the bacterial order *Rhodobacterales*, which play key roles in marine ecosystems and in bacterial-algal interactions. The authors identified three distinct types of flagellar gene clusters (FGCs), finding that the most common type, *fla1*, was likely inherited vertically from a common ancestor, whereas *fla2* and *fla3* appear to have spread through horizontal gene transfer. Using *Phaeobacter inhibens* as a model organism, they examined the structural and regulatory determinants of motility, identifying genes essential for movement, including four conserved genes of unknown function within the FGC (CP1–4) and the phosphorelay system CtrA-ChpT-CckA. Comparative analysis of CtrA regulation across different *Rhodobacterales* species suggests a conserved link between the CtrA phosphorelay and flagellar regulation, as well as its integration with other regulatory networks.

The manuscript is clearly and thoroughly written, with a high level of detail. Overall, the data analysis and the methodology appear robust and appropriately conducted.

Here some comments.

The main concern regarding this study is that the characterized Δ CP1–4 mutants were generated through transposon mutagenesis. This approach could cause a polar effect, meaning that the observed phenotype might result from the loss of function of downstream genes, whose activity could be partially or completely disrupted. Although complementation in trans was performed for each gene (Fig 3.C), leading to phenotype restoration, in most cases the complemented strains exhibited only partial recovery. It remains difficult to determine whether this partial phenotype is due to plasmid copy number effects or to a polar effect impacting downstream genes from the mutated target. None of the figures show wild-type control with the empty plasmid, nor the phenotype of the unmodified wild-type strain.

Regarding this same topic, genes CP2, 3, and 4 appear to belong to the same gene cluster, is that correct? If so, disrupting CP2 should affect the expression of downstream genes, including CP3 and CP4, right? If that's the case, complementation of Δ CP2 shouldn't show a phenotype, and neither should Δ CP3 due to the absence of CP4. How can this phenomenon be explained?

The CP1 gene appears to be located alone in the case of *P. inhibens* (the surrounding region is not shown). In this case, wouldn't the mutation have no polar effect? Could you confirm this?

On the other hand, mutation of genes related to the phosphorelay system affects the expression of genes involved in c-di-GMP metabolism. According to the results, both diguanylate cyclases and phosphodiesterases are impacted. Overall, how are the intracellular levels of c-di-GMP affected? Does this influence the bacterium's ability to form biofilms or adhere to surfaces? In this species, is the flagellum used as a mechanosensor, as observed in *Caulobacter*? Does the loss of function trigger the expression of genes encoding a polar adhesin (similar to the holdfast or the UPP (unipolar polysaccharide)? Do the transcriptomic data provide any indication of this?". The rosette structure observed in this bacterium may be related to this phenomenon.

Finally, no class III regulator (FlbD-like) was identified in the transposon screening. Is there no homolog present in the genome? Were no potential candidates detected in the transcriptomic analysis of the Δ ctrA mutant? In light of this, how is the regulatory hierarchy proposed?

Could the polar structure observed in Figure S4 and mentioned on line 257 be related to some type of polar adhesion phenotype? In several panels, cell-to-cell attachment by the poles can

be seen — is this type of interaction common in this bacterium? Is the structure associated with this phenotype known? Could it be the Tad pilus?

Minor comments:

Pag 7 line 187-189: in all before papers *fla1* in *C. sphaeroides* is referred as *fla2*, is this correct?, In that case it would be useful to clarify the nomenclature in some way to facilitate the comprehension

Pag 7 line 207-209: “A mutant of the putative flagellar biosynthesis repressor *flbT* was obtained from the sequenced transposon library. However, in accordance with its presumed role, it was still motile”. This statement may not be entirely accurate. FlbT has been reported to function either as an activator or a repressor in different bacterial species. In the present case, it appears to play a role similar to that observed in *Caulobacter*, where its mutation does not completely abolish swimming motility. Is the phenotype of the *flbT* mutant comparable to that of the wild type? Are there any differences in swimming halo diameter between the two strains? Additionally, are *fliC* mRNA levels altered in the *flbT* mutant? In *Caulobacter*, the *flbT* mutant accumulates flagellins. Are flagellin levels altered in this mutant strain? A straightforward, semi-quantitative approach to assess this would be to purify flagella from the culture supernatant and analyze them by SDS-PAGE.

Pag. 9 line 274: Is the mentioned *motA* copy functional or related to the flagellum? Is there a connection between the regulation of this gene and the other copy?

Response to reviewers' comments to author:

This MS focuses on flagellar motility in the bacterial order Rhodobacterales, which play key roles in marine ecosystems and in bacterial-algal interactions. The authors identified three distinct types of flagellar gene clusters (FGCs), finding that the most common type, *fla1*, was likely inherited vertically from a common ancestor, whereas *fla2* and *fla3* appear to have spread through horizontal gene transfer. Using *Phaeobacter inhibens* as a model organism, they examined the structural and regulatory determinants of motility, identifying genes essential for movement, including four conserved genes of unknown function within the FGC (CP1–4) and the phosphorelay system CtrA-ChpT-CckA. Comparative analysis of CtrA regulation across different Rhodobacterales species suggests a conserved link between the CtrA phosphorelay and flagellar regulation, as well as its integration with other regulatory networks.

The manuscript is clearly and thoroughly written, with a high level of detail. Overall, the data analysis and the methodology appear robust and appropriately conducted.

Answer: We would like to thank the Reviewer for the evaluation and appreciation of our work.

Here some comments.

The main concern regarding this study is that the characterized Δ CP1–4 mutants were generated through transposon mutagenesis. This approach could cause a polar effect, meaning that the observed phenotype might result from the loss of function of downstream genes, whose activity could be partially or completely disrupted. Although complementation in trans was performed for each gene (Fig 3.C), leading to phenotype restoration, in most cases the complemented strains exhibited only partial recovery. It remains difficult to determine whether this partial phenotype is due to plasmid copy number effects or to a polar effect impacting downstream genes from the mutated target. None of the figures show wild-type control with the empty plasmid, nor the phenotype of the unmodified wild-type strain.

Regarding this same topic, genes CP2, 3, and 4 appear to belong to the same gene cluster, is that correct? If so, disrupting CP2 should affect the expression of downstream genes, including CP3 and CP4, right? If that's the case, complementation of Δ CP2 shouldn't show a phenotype, and neither should Δ CP3 due to the absence of CP4. How can this phenomenon be explained?

Answer:

--- We agree that the control experiments are important. They have been done and are now included in the revised manuscript. Swimming motility of all mutants was compared with the *Phaeobacter inhibens* DSM 17395 wild type as a positive control on the same soft agar plate. Furthermore, the empty pBBR1-MCS5 vector without the construct, which was transformed into the transposon mutants, served as negative control. The complementation assay including the positive controls and the negative controls are now included as Supplementary Figure S3.

--- Our comparative genome analyses have shown that four conserved proteins (CP1-4) are universally present in *fla1*-type flagellar systems of *Rhodobacterales*, leading to the prediction that they may be essential for swimming motility. The prediction was supported by at least two independent transposon mutants for each of the genes and comparable results for *Epibacterium* sp. TM1040 (Belas et al. 2009). However, we were aware of potential polar effects due to the localization of CP2, CP3 and

CP4 in the *fliL*-CP2-CP3-*motA*-CP4 gene cluster. Accordingly, we (i) cloned the whole gene cluster including the promoter region into the broad host-range low copy-number vector pBBR1-MCS5 for the complementation of the CP4 transposon mutant. The motility assay with the construct resulted in restoration of the swimming phenotype, confirming our conclusion that CP4 is essential for flagellar motility. Furthermore, we (ii, iii) cloned the CP2 and CP3 gene with the strong promoter of the gentamicin resistance gene into pBBR1-MCS5. Transformation of these constructs into the immotile mutants also resulted in the restoration of swimming motility on soft agar plates. Accordingly, our experiments clearly showed that both genes, CP2 and CP3, are essential for motility mediated by the *fliA*-type flagellum of *P. inhibens* DSM 17395.

--- We agree that the low-copy number vector as well as the gentamicin promoter used for our complementation assays might have effects on the swimming behavior. However, complementation was intended to demonstrate the essential role of the four conserved proteins. Comparison with the wild type showed that all four complemented transposon mutants swim comparably well.

--- Given the successful complementation of the CP2 and CP3 transposon mutants, we can only speculate about downstream effects on the *fliL*-CP2-CP3-*motA*-CP4 gene cluster. Gene expression of the downstream genes might be induced by individual promoters, irrespective of gene clustering. Another explanation could be protein translation from a polycistronic mRNA, regardless of the integration of the EZ-Tn5™ transposon with a length of 2,001 nt.

--- For better traceability of our genetic complementation experiments, we now present all primers used for the cloning experiments in the **Supplementary Table S7**. We now further specify in the material and methods section (**line 507**): “The single CP1 gene and the whole CP4 containing operon were cloned together with the upstream intergenic region into the broad host range expression vector pBBR1-MCS5. For CP2 and CP3, the constitutively active gentamicin promoter was cloned in front of the ORF.”

The CP1 gene appears to be located alone in the case of *P. inhibens* (the surrounding region is not shown). In this case, wouldn't the mutation have no polar effect? Could you confirm this?

Answer: Yes, the putative transcriptional regulator downstream of the CP1 gene is located on the opposite strand. Therefore, a polar effect by the integration of the transposon is unlikely.

On the other hand, mutation of genes related to the phosphorelay system affects the expression of genes involved in c-di-GMP metabolism. According to the results, both diguanylate cyclases and phosphodiesterases are impacted. Overall, how are the intracellular levels of c-di-GMP affected? Does this influence the bacterium's ability to form biofilms or adhere to surfaces? In this species, is the flagellum used as a mechanosensor, as observed in *Caulobacter*? Does the loss of function trigger the expression of genes encoding a polar adhesin (similar to the holdfast or the UPP (unipolar polysaccharide))? Do the transcriptomic data provide any indication of this?". The rosette structure observed in this bacterium may be related to this phenomenon.

Answer: The role of c-di-GMP is indeed very interesting. However, we did not perform c-di-GMP quantification for *P. inhibens* in the current study. In a previous study on quorum sensing in *D. shibae*, we found that transcription of the two diguanylate cyclase and two phosphodiesterase genes present

(*D. shibae* has a very small number of c-di-GMP turnover genes compared to *P. inhibens* and others) is activated sequentially after addition of autoinducer signaling molecules and after the QS activated *ctrA* genes. The c-di-GMP levels rise after the addition but a drop was not observed, probably because the observation time was not long enough. Clearly, in both organisms control of this second messenger is linked to *ctrA* and motility. The extended set of turnover genes, not all of them under control of *CtrA* in *P. inhibens* could mean that additional processes are controlled by this second messenger. Analyzing c-di-GMP physiology would open a whole new chapter and would be beyond the scope of the current study. Here we conclude with the statement in **line 408**: “the *CtrA* core regulon of motility and cyclic-di-GMP turnover and its integration into the QS system are conserved in these two phylogenetically distant species of *Roseobacteraceae*.” We agree that further research on the role of c-di-GMP in attachment and biofilm formation or the role of flagella in surface sensing would be of great benefit and should be addressed in the future.

Only one polysaccharide related gene showed differential expression in the *ctrA* mutants. PGA1_c05480 is a polysaccharide deacetylase. The *C. vibrioides* holdfast polysaccharide deacetylase HfsH has been shown to modulate adhesive strength of the holdfast (<https://doi.org/10.1111/mmi.12199>). However, PGA1_c05480 does not show significant similarity to HfsH but to the allantoinase PuuE thus playing a different role than adherence. To our knowledge, the genes responsible for biosynthesis of the polar polysaccharides that mediate rosette formation of *P. inhibens* are unknown.

Finally, no class III regulator (FlbD-like) was identified in the transposon screening. Is there no homolog present in the genome? Were no potential candidates detected in the transcriptomic analysis of the Δ *ctrA* mutant? In light of this, how is the regulatory hierarchy proposed? Could the polar structure observed in Figure S4 and mentioned on line 257 be related to some type of polar adhesion phenotype? In several panels, cell-to-cell attachment by the poles can be seen — is this type of interaction common in this bacterium? Is the structure associated with this phenotype known? Could it be the Tad pilus?

Answer: Thank you for this question. Indeed, we found two close homologs of *Caulobacter vibrioides flbD* in the genome of *P. inhibens*, PGA1_c27610 (evalue = $4e^{-62}$) and PGA1_c14810 (evalue = $5e^{-53}$). Neither of these genes were differentially expressed in the *CtrA* phosphorelay mutants. Therefore, we cannot draw reliable conclusions about their potential integration into the regulatory hierarchy. However, the latter, PGA1_c14810 annotated as NtrX-like transcription factor in *P. inhibens*, was hit eleven times in the transposon motility screening. Therefore, we assume also a regulatory role like *flbD* for this gene. We modified the text accordingly and added the reference to *flbD* in *Caulobacter* (**line 221**).

We performed the electron microscopy mainly to monitor the presence or absence of flagella. While the observed structures are interesting, we think that our data is not strong enough to support further speculations about their origin or function. Attachment of *P. inhibens* to surfaces and other microorganisms is indeed common. *P. inhibens* forms rosette-like structures by attachment of several cells at their poles. This attachment is likely mediated by an N-acetylglucosamine-containing polysaccharide (see <https://doi.org/10.1371/journal.pone.0141300>). As emphasized above, we agree that further research in this direction is needed to elucidate the swim-or-stick lifestyle of *P. inhibens*. It

is, however, beyond the scope of the current manuscript about the archetypal flagellar system of *Rhodobacterales*.

Minor comments:

Pag 7 line 187-189: in all before papers *fla1* in *C. sphaeroides* is referred as *fla2*, is this correct?

In that case it would be useful to clarify the nomenclature in some way to facilitate the

Comprehension

Answer: Thank you for noting. You are correct. We clarified the nomenclature in the discussion (line 447): “Note that in this bacterium the *fla1* and *fla3*-type FGC were originally named in order of their discovery, *fla1* and *fla2*, respectively.”

Pag 7 line 207-209: “A mutant of the putative flagellar biosynthesis repressor *flbT* was obtained from the sequenced transposon library. However, in accordance with its presumed role, it was still motile”. This statement may not be entirely accurate. *FlbT* has been reported to function either as an activator or a repressor in different bacterial species. In the present case, it appears to play a role similar to that observed in *Caulobacter*, where its mutation does not completely abolish swimming motility. Is the phenotype of the *flbT* mutant comparable to that of the wild type? Are there any differences in swimming halo diameter between the two strains? Additionally, are *fliC* mRNA levels altered in the *flbT* mutant? In *Caulobacter*, the *flbT* mutant accumulates flagellins. Are flagellin levels altered in this mutant strain? A straightforward, semi-quantitative approach to assess this would be to purify flagella from the culture supernatant and analyze them by SDS-PAGE.

Answer: Thank you for these questions. The phenotype of the *flbT* mutant of *P. inhibens* DSM 17395 is comparable to the wild type with no difference in the swimming diameter (see the new Supplementary Figure S3A). This is in accordance with the presumed role as a repressor. We did not perform transcriptomics of the *flbT* mutants, and although this is interesting, an additional focus on *FlbT* function is not required for the understanding of the present manuscript.

Pag. 9 line 274: Is the mentioned *motA* copy functional or related to the flagellum? Is there a connection between the regulation of this gene and the other copy?

Answer: Thank you for this question. The *motA2* gene outside of the FGC showed a different expression pattern compared to *motA1* inside the FGC. It was stronger expressed in the *ctrA* and *cckA*, but not the *chpT* mutant. *MotA1* was downregulated in all three mutants. Interestingly, the *motA2* expression pattern is similar to the siderophore genes. This similarity is now highlighted in the text (line 298).

Re: mSystems00419-25R1 (Structural and regulatory determinants of flagellar motility in Rhodobacterales - The archetypal flagellum of *Phaeobacter inhibens* DSM 17395)

Dear Dr. Jürgen Tomasch:

Your manuscript has been accepted, and I am forwarding it to the ASM production staff for publication. Your paper will first be checked to make sure all elements meet the technical requirements. ASM staff will contact you if anything needs to be revised before copyediting and production can begin. Otherwise, you will be notified when your proofs are ready to be viewed.

Sincerely,
Maria Scrascia
Editor
mSystems

Reviewer #1 (Comments for the Author):

The authors have addressed all the suggestions in a thorough and effective manner. All comments were taken into account and have been convincingly resolved. The manuscript has significantly improved, and I have no further objections regarding the work.